# Cranial Mandibular Fibrosis Syndrome in Adult Farmed Rainbow Trout *Oncorhynchus mykiss*

**DOI:** 10.3390/pathogens10050542

**Published:** 2021-04-30

**Authors:** Irene Cano, John Worswick, Brian Mulhearn, Matt Green, Stephen W. Feist, Morag Clinton

**Affiliations:** 1International Centre of Excellence for Aquatic Animal Health, Cefas Weymouth Laboratory, Weymouth, Dorset DT4 8UB, UK; john.worswick@cefas.co.uk (J.W.); brian.mulhearn@icloud.com (B.M.); matthew.green@cefas.co.uk (M.G.); oie.cceaad@cefas.co.uk (S.W.F.); morag.clinton@alaska.edu (M.C.); 2Department of Veterinary Medicine, Institute of Arctic Health, University of Alaska Fairbanks, Fairbanks, AK 99775, USA

**Keywords:** rainbow trout, mandibular fibrosis, dermatitis, PKD, *Tetracapsuloides bryosalmonae*, TGF-b, immunoglobulaemia, lysozyme, calcinosis

## Abstract

An unusual condition affecting market size rainbow trout was investigated. This condition was prevalent for several years at low levels but affected a large proportion of stock during 2018 and 2019. Chronic fibrosis affecting cranial tissues and the jaw was observed in samples collected in 2018. A larger sampling was then conducted in 2019 to investigate the presence of an infectious agent(s). An extensive inflammatory response in the mandibular region was the main finding, however infectious agents in the lesions were not identified through classical virology and bacteriology analysis. *Tetracapsuloides bryosalmonae* infection, calcinosis, and a Gram-positive bacterial infection of a single fish cardiac tissue was observed, however, a correlation of these pathologies and the cranial mandibular fibrosis (CMF) syndrome was not established. The gene expression of a panel of 16 immune-related genes was studied. Among these, *tgf-b*, *sIgM*, *il11*, *hspa*, and the antimicrobial peptides *lys* and *cath1* were up-regulated in jaw sections of CMF-affected fish, showing a strong positive correlation with the severity of the lesions. Idiopathic chronic fibrosis with the activation of the Tfg-B pathway and local hyper-immunoglobulaemia was therefore diagnosed. Initiating factors and causative agent(s) (biotic or abiotic) of CMF remain, at present, unclear.

## 1. Introduction

Disease problems in aquaculture are estimated to cost the global industry US$ 6 billion per year [1]. Biosecurity programs mainly focus on preventing, controlling, and eradicating infectious diseases to meet governmental regulations’ objectives and align with FAO’s Blue Growth Initiative goals for sustainable aquaculture development and food security [2]. However, the cost estimate for many diseases is lacking, particularly those not included in the ‘top’ health conditions impacting aquaculture. Less well-understood disorders can go unspotted, undiagnosed, and under-reported [3]. 

Among those under-reported health conditions, multiple disorders related directly or indirectly to water pollution are known to have severe animal welfare implications and economic impact on aquaculture industries. Hyperplasia, papilloma, fin and tail rot, gill diseases, liver damage, neoplasia, and ulceration have been identified in fish due to immunosuppression, reduced metabolism, and gill and skin damage induced by pollutant exposure [4]. Those disorders can occur long after the contamination episode or as a chronic effect of long-term exposure. Thus, the link between pollutants and disease is often difficult to prove. 

Skeletal and developmental disorders have also been documented in economically important cultured fish species. Some of the most commonly reported skeletal disorders include lower jaw deformity (LJD) affecting farmed triploid Atlantic salmon (*Salmo salar* Linnaeus, 1758), described as a downward curvature of the lower jaw involving dentary and glossohyal bones [5]; short opercula and gill filament deformity syndrome (GFD), defined as an absence of primary gill filaments in Atlantic salmon [6]; and abnormal swimbladder and lordosis conditions, characterized by a V-shaped spine in sea bass (*Dicentrarchus labrax* L.) and gilthead seabream (*Sparus auratus* L.) [7]. These deformities have been traditionally associated with husbandry factors, rearing temperatures, and/or ploidy [8,9].

Economic losses due to diseases of an unknown agent are particularly of concern to aquaculture producers, and are considered to go often under-reported [3]. Among conditions of unclear etiology described in salmonids, puffy skin disease (PSD) affects both diploid and triploid rainbow trout (*Oncorhynchus mykiss* L.). PSD has been reported so far mainly in the United Kingdom (UK). PSD is characterised by epidermal hyperplasia and emaciation of fish, resulting in a downgrading of carcasses at slaughter [10,11]. Although a plethora of parasites has been associated with PSD-impacted fish, among them *Ichthyophthirious multifiliis* (Fouquet 1876) and *Ichthyobodo necator* (Henneguy 1883), the actual causative agent(s) of PSD remain unclear [12]. Another infectious skin disease affecting rainbow trout is red mark syndrome (RMS), also known as cold water strawberry disease (CWSD). This condition has been reported in the UK, the USA, Switzerland, and Austria [13,14]. RMS causes skin lesions characterised by lichenoid dermatitis and infiltration of mononuclear cells, with pathology observed mainly in the dermis [15]. *Midichloria*-like organism DNA, a bacteria belonging to Rickettsiales, has been associated with RMS lesions [16]. 

Other emerging diseases affecting wild salmonid populations include red vent syndrome (RVS), red skin disease (RSD), and ulcerative dermal necrosis (UDN). RVS, reported in wild Atlantic salmon in the UK, Ireland, Norway, and Canada [17], is characterised by red, swollen vents. Although an association of *Anisakis simplex* with these lesions has been noted, environmental changes have also been considered as either directly or indirectly linked to the syndrome (i.e., increasing parasite load) [18]. RSD was first reported in Sweden, and more recently in additional European countries [19]. RSD is associated with haemorrhage and necrosis in the skin, with concurrent opportunistic fungal infections [20]. UDN is a condition described in wild salmonid populations of Baltic waters. The disease is characterised by cytolytic necrosis of epidermal Malpighian cells of the head. Circulatory failure and death of UDN-affected fish results from the combined negative factors of skin ulcerations and infection by opportunistic pathogens, including oomycete *Saprolegnia diclina* and secondary bacteria [21]. 

In non-salmonid farmed fish, several emerging skin diseases have also been reported. In fish farms in the Mediterranean sea, red spot disease in sole (*Solea* spp.), ulcerative dermatitis in dusky grouper (*Epinephelus marginatus* Lowe 1834), PSD-like in seabass, and petechial rash (PR) in gilthead sea bream and sea bass have all been described [22].

Predisposing factors for many conditions in salmonids appear complex, particularly those of apparently mixed aetiology. Therefore, understanding the environmental and host factors that influence disease is a focus of research in cultured salmonids. The present study describes a novel syndrome affecting size market rainbow trout observed, so far, on a single farm in the UK. The gross pathology, histopathology and microbiology examinations are described alongside the gene expression pattern of several immune-related genes. These findings provide a case definition that could help identify this condition in other farms and epidemiological studies. Along with key diagnostic indicators, plausible causative agent(s) of biotic and/or abiotic origin are discussed.

## 2. Results

### 2.1. Historical Presentation of Cranial Mandibular Fibrosis Syndrome (CMF)

In 2018, the Fish Health Inspectorate (FHI) in England and Wales reported a fish farm in South England with an unusually high number of adult rainbow trout (>300 g) presenting with apparent pathology in the jaw and head, characterized by swelling and reddening of the area. The condition, first noticed around 2007 by the fish farmer at a low level, was reported to affect a large proportion of stock during 2018 and 2019. This farm (called site A) produced triploid, market-size animals intended for restocking. The condition became observable once the trout have been on site for 4–5 months. As the trout grew and the deformity became more apparent, fish struggled to eat and eventually to respirate. No previous major health issues were reported in recent years from this farm aside from seasonal *Tetracapsuloides bryosalmonae* infection. The impacted site received juvenile trout from a nearby hatchery. This same hatchery provided trout of identical genetic background (same population) to a second site (called site B). Both farm sites A and B were located on the same stream. However, the same condition was never observed in site B or co-habiting brown trout (*Salmo trutta* L.) stocked on the affected site (site A). A sewage treatment plant outlet and a small industrial estate are located between the two farms, with the affected farm downstream. 

### 2.2. Gross Pathology Description

In 2018 (Spring and Autumn samplings) only fish showing gross pathology were examined. Clinical signs were a progressive ‘cheek’/jaw deformity in the head. Tissue change was apparent caudal to the suborbital structures and extending to the opercular gill covering, involving the structures of the head, mandibular region, branchiostegal rays (opercular folds) and rostral opercular structures. Tissue in afflicted fish appeared swollen and erythematous (reddened). Presenting most frequently unilaterally, tissue changes lead to a complete fixed gaping (Figure 1a). Under-opercular deformity and a dense pink swelling without clear margins were present in lower opercular folds and the head (Figure 1b). Eye damage and occasional spinal deformity/ breakage were noted within the population. Internally, the organs of sampled fish looked macroscopically normal.

From fish sampled in 2019, similar swelling and unilateral deformities of the region of the jaw were also observed, however, swelling in other cranial tissues was less evident. 

Fish samples from Site A in 2019 were grouped according to the lesions’ gross clinical severity as severe (Figure 1c,d), moderate, mild, and no lesions (Figure 1e,f) based on the degree of swelling. Affected fish were reported to demonstrate inappetence and altered swimming behaviour during all site visits. Fish sampled from the nearby Site B (not shown) did not show any apparent lesions or behavioural alterations.

### 2.3. Histopathology Findings 

The samples received in Spring 2018 showed lesions on the operculum and jaw. Histopathology of this swollen tissue region demonstrated extensive increased cellularity of the dermis and surrounding the cartilaginous structures (Figure 2a). A mixed immune cell infiltrate primarily of histiocytic and lymphocytic cell types was present within the dermal tissue constituting a diffuse chronic dermatitis (Figure 2b). This increased immune cellularity was observed too within neural structures and infiltrating underlying adipose and muscular tissue. Some regions of the gill showed epithelial hyperplasia and an increased number of eosinophil granule cells (Figure 2c), however these gill changes were considered likely due to gill fouling from debris and environmental microorganisms, including the *Saprolegnia* and unidentified bacteria observed. In internal visceral tissues, some individuals demonstrated marked perivasculitis as well as multifocal early granulomatous changes in the liver and kidney (surrounding collecting ducts), with some glomerular involvement. Increased numbers of circulating monocytes were noted. Two hearts presented with mild epicarditis (Table 1).

Further samples collected in September 2018 showed similar dermatitis affecting the stratum compactum and underlying adipose and muscle tissue of the affected jaw sections, with immune cell infiltration of connective tissues, including around vasculature. A prominent histiocyte presence was observed in fish sampled at this time, alongside other immunological cell types, including lymphocytes. Infectious agents were not observed in the jaw tissues. Lymphocytic infiltration of the lamina propria of the stomach (1 fish out of 10) and lymphocytic inflammatory foci within the parenchyma of the liver (3 fish out of 10) was observed. Liver pathology was considered to be associated with the presence of *Tetracapsuloides bryosalmonae*, the causative agent of proliferative kidney disease (PKD). Pre-sporogonic *T. bryosalmonae* were observed in 5 out of 10 fish renal, hepatic, and splenic tissues. Focal inflammatory cell infiltration of the myocardium of a single fish was also observed, associated with a Gram-positive bacterial infection (Figure 2d). In this fish, an inflammatory cell aggregate was observed in the kidney with prominent neutrophil presence and cellular necrosis, alongside nephrocalcinosis. 

The histopathology of the rainbow trout collected in March of 2019 was consistent with the previous samplings. Extensive inflammation within the connective tissues of the dermis was present in conjunction with necrosis and disruption of normal tissue structure. These fish also demonstrated a thickened stratum compactum with increased connective tissue presence (fibrosis) in underlying tissue structures. Fibroblast cell type presence was noted accompanying this marked dermal inflammation (dermatitis) (Figure 3b,d,f). This extensive inflammatory infiltration was observed in affected fish throughout the stratum compactum of the skin, with inflammatory infiltration extending to underlying adipose, skeletal muscle, and periosteal connective tissue structures. The underlying musculature involvement was isolated primarily to intramuscular septa, with only single foci of necrotic change to skeletal myocytes themselves (Figure 3h). 

In fish showing severe lesions, perivascular inflammatory infiltration in the epineurium and tissue of neural fibers were prominent (Figure 4b). Lesions were characterised by immunological infiltration of dermal tissues by immune-cell types, with a high noted presence of histiocytes and lymphocytes (Figure 4c), although other immunological cell types were observed. Sporadic presence of haemorrhagic lesions was noted within the dermis, however, they do not appear to be a consistent feature of pathology. Similarly, multinucleated giant cells (Figure 4d) were noted infrequently in severely impacted fish. Mild to moderate perivascular infiltration was observed in the circulatory structures of the subcutaneous tissue of most sampled fish (Figure 4e). No infectious agents were observed in these tissue sections. Epithelial tissue was largely unaffected, except for a single foci of epidermal inflammatory infiltration (predominantly of lymphocyte cells), observed in a single individual. 

In some affected fish, focal nephrocalcinosis and calcified nodules in the lamina propria and muscularis mucosa of the stomach (Figure 4f,g) were also observed. Other incidental pathologies included focal lesions in the kidney and liver associated with PKD. Variable presence of *T. bryosalmonae* cells was noted (Figure 4h). Although mild epithelial hyperplasia and lamellar fusions were noted in gill tissue of multiple fish, these changes were also present within concurrently assessed apparently healthy fish from the same population. No additional pathology of note was observed within these fish. Microscopic observation of blood smears did not show any abnormalities (data not shown). Skeletal elements and ocular tissue were not assessed. 

Histopathological changes observed in fish presenting with more clinically severe swelling and erythema of the head included extensive and marked inflammatory infiltration of dermal tissue layers by histiocytic and lymphocytic cell types. Marked infiltration of underlying osteal connective tissues and musculature was observed as well in fish considered to have a severe clinical presentation. 

### 2.4. Microbiology Findings 

No colonies were obtained from kidney swabs or swabs taken from aseptic incisions in the jaw. Viral pathogens were not isolated by cell culture, although cytopathic effects (CPEs) were observed in RTG-2 cells inoculated with a pool of internal organs from diseased fish. The development of CPEs was attributed to presumed *T. bryosalmonae* cells in the culture derived from the kidney and/or spleen homogenates. 

### 2.5. Molecular Tests 

To rule out the presence of an underlying viral infection, the supernatant of cells showing CPEs were subjected to an array of molecular tests. The supernatants tested negative in an IPNV antigen (ag) ELISA. RT-PCR analysis using generic primers for species in the genera *Novirhabdovirus*, *Aquabirnavirus*, *Iridovirus*, and *Vesiculovirus* also yielded either no amplification or unspecific amplicons.

### 2.6. Next-Generation Sequencing

To assess for other possible viral agents or pathogens in association with the observed CPEs, DNA and RNA extracted from the inoculated cells were analysed using a MiSeq sequencer. Sequences obtained from the following preparations: total DNA (6,642,000 reads), total RNA (10,832 reads), viral single-stranded (ss)RNA (96,000 reads), viral DNA (160,216 reads), and viral double-stranded (ds)RNA (3,716,382 reads) were *the novo* assembled and mapped against reference organisms (viral hemorrhagic septicemia virus (VHSV), infectious pancreatic necrosis virus (IPNV), epizootic haematopoietic necrosis virus (EHNV) and salmonid alphavirus 2 (SAV2). None of the consensus sequences aligned with those reference viral genomes or any other known pathogens when blasted against the nucleotide NCBI database.

### 2.7. Host Response

Following clinical observations and histopathology, the fish collected on the affected site (site A-2019) were grouped according to the severity of the lesions in severe (*n* = 8), moderate (*n* = 4), mild (*n* = 4), and non-lesions (*n* = 7). Fish collected from an unaffected site (site B) of the same genetic background were considered negative controls (*n* = 7). Those groups were then used for gene expression analysis. 

For each sampled fish, a panel of 16 immune-related genes was analysed in jaw and kidney sections. Hierarchical clustering of the gene expression patterns in jaw sections showed that fish displaying severe and moderate lesions clustered together and that that cluster branched with the group displaying mild lesions. Fish without lesions and control fish formed an independent cluster (Figure 5a). Similarly, the gene expression patterns in kidney sections showed a cluster formed by the severe and moderate lesion groups and a cluster formed by mild lesions and no-lesions groups. Control fish showed an independent cluster (Figure 5b).

Of those genes, the *transforming growth factor-beta* (*tgf-b*), *interleukin 10* (*il10*), *immunoglobulin M heavy chain* (*igm*), *lysozyme II* (*lyz2*), and the *heat shock 70 kDa protein 12A*-like (*hspa12a*) were significantly up-regulated (*p* < 0.05) in fish with cranial mandibular fibrosis lesions when compared with control fish from site B. The antimicrobial peptides *cathelicidin 1* (*cath1*) and *interleukin 11* (*il11*) were also found to be significantly up-regulated in fish with severe lesions only. The gene expression of *cluster of differentiation 8* (*cd8a*) and *viperin* (*vig1*) were observed to be down-regulated in fish with severe lesions when compared with the no-lesions group collected in the affected site. 

In the kidney, *il11* was observed up-regulated in severe and moderate lesions compared with the no-lesion group. While the gene expression of the *cd8a*, *interleukin 4/13a* (*il4/13a*), *t-bet* (*tbx21*), *GATA-binding protein 3* (*gata3*), and *il10* were observed down-regulated in the severe and/or moderate lesions group when compared either with control fish or fish not affected sampled in the same site (Figure 5 and Appendix A). 

### 2.8. Correlation of PKD with Cranial Mandibular Fibrosis Syndrome

To investigate a possible association of chronic PKD with the appearance of cranial mandibular fibrosis lesions, a novel Taqman qPCR was designed and tested on the kidney samples. Coinciding with the histological observations, the *T. bryosalmonae 60S ribosomal protein l18* (*rpl18*) mRNA was detected solely in the affected site in a limited number of specimens showing severe lesions (in 3 fish out of 8) and in the group without lesions (in 1 fish out of 7). Still, it was not detected in the groups showing mild or moderate lesions (Appendix A). 

## 3. Discussion

Many non-OIE listed diseases affecting farmed fish likely go under or unreported globally, either due to a lack of current knowledge or no follow-through with diagnostics. Contamination events and novel agents or emerging infectious diseases (EIDs) are particularly challenging to diagnose, including outbreaks such as reported here with small geographical distribution.

This study describes a novel condition termed cranial mandibular fibrosis syndrome (CMF) affecting adult rainbow trout farmed in the UK. The detailed description of this syndrome based on the clinical presentation, histological examination and host response will help identify similar cases to aid in future reporting, which may ultimately reveal a more widespread extent of this syndrome. 

Consistently, affected fish showed a progressive jaw deformity, typically affecting only one side of the head. In severe cases, this deformity impaired gaping and mobility of the jaw. Histopathology revealed extensive inflammation surrounding the cartilage and infiltrated muscular and connective tissues. Multinucleated giant cells, formed by the fusion of macrophages [23], were sporadically observed in foci of inflammation. Their nuclear arrangement at one pole suggests they were of Langhans’ giant cell phenotype [24]. These cells secrete interleukins, promoting inflammatory processes associated with chronic inflammation and granulomatous conditions [24,25]. Inflammatory cells constituting the bulk of the cell infiltration included histiocytes and lymphocytes; mononuclear cell types that predominate in chronic inflammatory skin conditions [26].

The intense inflammation observed surrounding neural tissue potentially might be associated with neurological deficits and altered control of the jaw [27]. A sensible question might be whether nerve damage could ultimately cause the functional impairments seen in fish. The condition trigeminal nerve neuritis, which results in paralysis of the muscles of the jaw, can manifest as a “dropped jaw” [28]. This pathology has been described in animals and humans [29,30]. Although the outcome is the same (inability to eat) the affected fish, in this case, showed a “closed jaw” instead of a “dropped jaw”. Similarly, inflammatory myopathies can impact joint function and lead to impaired movement [31], such as observed in these fish, and so it is unclear from the pathology what specifically has led to the functional deficits. 

The fact that new cohorts consistently developed similar pathology 4–5 months from being introduced to the site suggests either the involvement of an infectious pathogen or perhaps exposure to a contaminant. Some histologically similar dermal changes are reported in trout with an extreme host response to the condition RMS, associated with a rickettsial infection [16]. However, typically RMS is characterized by the appearance of focal or multifocal skin lesions on the flanks and the ventral or dorsal surfaces with involvement of epithelial tissue layers [15], and so far has not been described as affecting the jaw. Histologically, RMS also presents with osteoclastic resorption of scales and is generally reported with extensive lymphocyte infiltration into the subdermal adipose tissue and the connective tissue surrounding the muscle bundles [32]. CMF differs from RMS in that the lymphocytic infiltration is more extensive, affecting deeper tissues, and appears together with chronic fibrosis in dermal and sub-dermal tissues. The clinical presentation and histological features of CMF are also distinguishable from PSD. PSD is a transmissible disease in rainbow trout for which the causative agent remains unknown, characterised by changes predominantly to the epidermal layer of the skin [10,12]. The affected farm did not have historical records of RMS or other skin conditions such as PSD. 

Myocarditis was observed in some CMF affected fish sampled in 2018 but not in 2019. From those fish showing mild heart inflammation, a single fish had a focal Gram-positive bacterial infection in the heart. However, due to the single incidence of this finding, further molecular analyses were not conducted to identify the bacterium. An early response to bacterial infections does involve significant numbers of neutrophils and other immune cells [33]. Among the most common Gram-positive bacteria infecting salmonids is *Renibacterium salmoninarum*, the causative bacterial kidney disease (BKD) agent [34]. This bacterium grows intracellularly in the phagocytic cells of the fish, and its presence is typically associated with a granulomatous host response [34]. Typical BKD clinical signs, when present, include skin darkening, distended abdomens, blood-filled blisters on the flanks, bruising around the vent, internal granulomas, and petechial hemorrhages around the lateral line [34]. None of these signs were observed in any of the CMF fish sampled. Other pathogenic Gram-positive bacteria described in rainbow trout belong to the genus Carnobacterium, typically associated with kidney disease and external ulcerations [35]. It has also been observed in fish displaying histopathological changes characteristic of nephrocalcinosis [35]. The low prevalence of the bacteria infection in the trout within this study showing CMF syndrome (3.3%) suggests a low correlation among pathologies. Future samplings of CMF fish might offer opportunities to identify any other concurrent infections and their role in the development of the CMF pathology. 

Other secondary infections observed in fish with CMF were saprolegniosis, observed in fish sampled in Spring 2018, and PKD observed in samples from 2018 and 2019. *Saprolegnia* species are oomycetes that grow as patches of mycelia visible on the surface of the fish, often around the head, gills, tail, and fins [36]. The infection primarily affects the epidermal tissue, causing epithelial hyperplasia, eosinophilia, inflammation, and in severe cases, necrosis [37]. The distinct features of CMF, failure to identify hyphae within histological sections, and low prevalence of saprolegnia in the CMF affected fish suggest that the appearance of CMF was not related to saprolegniosis. On the other hand, the affected farm had extensive historical records of PKD infection. Some CMF fish demonstrated *T. bryosalmonae* parasites in the liver and kidney, accompanied by the multifocal inflammation and severe immunological response typical of PKD [38]. The presence of this myxozoan parasite was further confirmed by Taqman qPCR in the kidney of fish showing PKD-consistent pathology. This parasite is usually found in visceral organs but never has been described to our knowledge in bony or dermal tissues, as opposed to its relative *Myxobolus cerebralis*, the agent of whirling disease [39]. *M. cerebralis* has been shown to replicate in the ventral calvarium and gill arches of rainbow trout and other salmonids, causing multifocal cartilage necrosis and leukocyte infiltration [40]. *M. cerebralis* was however not detected in any of the CMF-affected fish, and the farm did not have historical records of whirling disease. 

Inflammatory processes can manifest in regions of skeletal muscle distant from the site of a stressor. Indeed, gilthead seabream (*Sparus aurata* L.) have been shown to respond to intraperitoneal administration of lipopolysaccharides (LPS) with a response that includes an inflammatory response in red skeletal muscle [41]. To investigate whether PKD infection might trigger an inflammatory response in tissues outside of the immune system (i.e., skeletal muscle of the jaw), a correlation of PKD with the CMF severity was evaluated. PKD was detected only in 37.5% of fish showing CMF severe lesions and 14.2% of fish without lesions sampled in the affected farm. PKD was, however, not seen in fish showing mild and moderate CMF lesions; therefore, a significant correlation of PKD with CMF was not established for the limited number of sampled fish. Unfortunately, altered farm (site A) activity in 2020 prevented a planned horizontal study from following a naïve cohort from on-site introduction to the development of CMF. Any correlation of CMF with PKD or other diseases, whether through direct pathogen action or impaired host immunological status for susceptibility to disease, was therefore no fully explored [42]. 

Noticeably, a pathological change observed in fish presenting with CMF was mild calcinosis of visceral organs (kidney and lamina propria and submucosal structures of gastro-intestinal tissue). Calcinosis is a condition in which calcium and other minerals precipitate within the distal renal tubules and collecting ducts (known as nephrocalcinosis); however, it can also occur in other soft tissues [43]. In mammals, calcium regulation is a complex process involving parathyroid hormone (PTH), vitamin D metabolites, and calcitonin [44]. In fish, calcium regulation is predominantly mediated by the hypocalcemic hormone STC-1 [45]. The corpuscles of Stannius are teleost-specific endocrine glands associated with the kidneys, which synthesize and secrete STC-1 [46]. Nephrocalcinosis in rainbow trout (*Salmo gairdneri*) have been previously documented concurrently to muscle lesions during incidences of severe renal damage [45]. Water quality and/or diet deficiencies can cause calcinosis, in particular, high levels of carbon dioxide in water, a diet low in minerals as magnesium, and toxicity to selenium and arsenic [47,48,49]. In the affected farm, the diet and fish stock density were considered within normal parameters, however, the water quality was not measured. Mild cases of calcinosis usually do not present a threat to the animal and may be considered incidental findings, however, in severely impacted animals, it can lead to a poor metabolic efficiency [49]. In the present study, mild calcinosis observed in CMF fish does not support a direct correlation across pathologies in the limited number of fish analysed.

In the absence of a pathogen identified on the jaw lesions, a possible connection of the syndrome with water contamination event(s) was also considered. Moreover, as the condition developed in new stock 4–5 months following introduction on-site, a chronic response to a toxin or irritant after repetitive exposures might be indicated [50]. Indeed, the location of a sewage treatment plant outlet and a small industrial estate upstream of the affected site could affect the water quality if the water discharged was not adequately treated. Fish exposed to contaminated waters can bioaccumulate copper, lead, cadmium, nickel, and zinc, among other heavy metals [51,52]. In general, gill tissues are most affected by heavy metal exposure, while bioaccumulation typically occurs in the kidney and liver [53]. Histological examination of the liver of CMF-affected fish did not show changes classically associated with heavy metals accumulation or other toxin exposure. Similarly, the CMF condition was not observed in brown trout on site. Lamentably, the concentration of heavy metals in the water, sediments, or in the affected fish was not measured, therefore a possible link between the CMF syndrome and contamination event(s) is purely speculative.

To investigate the host immune response involved in the chronic inflammation observed in CMF affected fish, the gene expression of a selection of immune-related genes was analysed. This array included cytokines, transcription factors, antimicrobial peptides (AMPs), immunoglobulin, complement, and lysozyme. These genes were used as biomarkers to reveal the cause of the inflammation and possible pathogen-driven T-cell polarization [54,55].

TGF-B has potent regulatory and inflammatory activity [56]. In particular, Treg polarization and over-regulation of TGF-B and its downstream signaling pathways strongly induce fibrosis [57]. The teleost homologous *tgf-b* has been characterized in several fish species [58,59]. In the present study, the synergetic up-regulation of the effector cytokines *tgf-b* and *il10* in CMF lesions suggested a humoral immune response, mediated by the activation of regulatory T (Treg) cells towards a controlled inflammatory response [60]. Our analysis also showed a significant up-regulation of the rainbow trout *il11*. Indeed it has been demonstrated that IL11 is a potent pro-inflammatory fibroblast activator and induces cell proliferation [61]. Moreover, the up-regulation of *igm* in CMF lesions denotes a chronic inflammation as shown in a mice skin model, where resident plasma cells secreted a high amount of IgM during chronic skin inflammation [62]. 

The up-regulation of *lyz2* and *cath1* in CMF lesions correlate with the high number of macrophages and leukocytes infiltration noted in the histology observations. Teleost cathelicidins have been identified in various fish species [63], showing a broad-spectrum activity against Gram-negative bacteria, Gram-positive bacteria, and fungi [64]. These AMPs are also potent modulators of the fish innate immune system in response to pathogens or other exogenous molecules to different pathogen-associated molecular patterns (PAMPs) [63,64]. In particular, a positive effect of AMPs in the expression levels of pro-inflammatory cytokines in macrophages has been shown for trout [65]. AMPs and other immunostimulants promote the activation of phagocytes, lymphocytes, complement, lysozyme, and a plethora of cytokines [55]. 

The recruitment of neutrophils, macrophages, T cells, B lymphocytes, eosinophil, and basophil granulocytes occur during chronic inflammation. These cells release a wide range of inflammatory cytokines, growth factors, and TGF-B [66]. Those cytokines then promote an excessive extracellular matrix production of interstitial fibroblasts, producing fibrosis [67]. Interestingly, the up-regulation of *hspa12a* in CMF lesions denotes the role of immune proteins on the pathogenesis of organ fibrosis [68]. Overall, the host response to CMF showed a marker pattern of fibrosis with polarization towards Treg cells, as opposed to other skin conditions RMS and PSD that lead a T helper (Th)1 and Treg type response in the case of RMS [32], or a Th17 type response in the case of PSD [69].

Significant changes in the kidney immune response of CMF affected fish were also measured. The site has historical reports of seasonal PKD infection. Although fish were sampled in March, when the prevalence of PKD is typically expected to be low (higher prevalence and intensity of the infection naturally occurs in Summer-Autumn when the water temperate is higher [70]), infection of *T. bryosalmonae* was detected in some of the fish sampled with a higher PKD incidence within the CMF-severe group. The affected site was also close to a sewage treatment plant. Although it was not the aim of this study to examine the effect of the sewage treatment plant and potential environmental contaminants on the health status of the animals, and thus water parameters were not measured, it has been shown that wastewater treatment plant effluent can significantly correlate with the prevalence and PKD infection intensity [71], likely caused by a local increase of the water temperature. Thus, collectively the data suggest that the general down-regulation of Th1 (*tbx21*) and Th2 (*gata3*) transcription factors, inflammatory cytokines (*il4* and *il10*), and cd8 (a cellular marker for cytotoxic T cells) in the kidney were attributed to a *T. bryosalmonae* chronic infection. Indeed, it has been shown that myxozoans *T. bryosalmonae* and *M. cerebralis* cause immunosuppression during parasites’ development [72]. 

As a summary, this study describes the pathological changes observed in the jaw tissues of market size rainbow trout impacted by CMF. CMF is characterized by idiopathic chronic inflammatory fibrosis that prevents gaping and mobility of the jaw, ultimately preventing feeding and, in severe cases, breathing. The host immune response in the affected tissues showed a Treg polarization pattern with the up-regulation of pro-inflammatory and pro-fibrosis cytokines and local hyper-immunoglobulaemia. Although the causative agent(s) (biotic or abiotic) of CMF remain unknown, chronic PKD infection and calcinocis of other tissues were highly prevalent in the affected farm. This case study represents the first description of this condition, may aid in the diagnostic of similar cases in the future.

## 4. Materials and Methods 

### 4.1. Ethical Statement 

All the animals used in this study were sampled as a result of official disease investigations. These animals were not subjected to a regulated procedure. Fish were euthanised humanely according to the UK Home Office procedures in compliance with the Animals (Scientific Procedures) Act 1986 Amendment Regulations 2012.

### 4.2. Field Sample Collection and Histology

Four diseased fish collected in the affected farm (site A) were received from a private consultant in 2018 for a histopathology assessment. Site A was then visited by the FHI in September 2018. Cultured fish were examined on-site, and ten rainbow trouts showing clinical signs were sampled for an initial pathology assessment. Site A was revisited in March 2019. On this occasion, 23 rainbow trouts were collected and brought to the laboratory for further pathology and microbiology assessments. In addition, 7 rainbow trouts of the same genetic background were collected from a nearby unaffected site (site B) as negative controls. All the sampled fish were diploid females weighing >300 g. 

Based on the gross pathology of the mandibular lesions, sampled fish in 2019 were categorised in a clinical severity index as severe (clear signs of jaw inflammation and swelling), moderate (apparent jaw inflammation but not swelling), mild (jaw redness and some indication of inflammation), or unaffected (no apparent clinical symptoms). This categorization aimed to include the study samples from the early (mild) and advance (severe) phases of the disease.

For fish sampled in 2019, blood smears were collected from the caudal vein, air-dried, and stained by the Giemsa method following standard protocols [73]. Post-mortem examination of internal organs for gross pathology was performed. Immediately after death, a section of the jaw was taken as a transverse excision to obtain a thin portion extending from lateral epidermal tissue to approximately central depth in a region just distal to the ocular margin (Appendix A). This allowed assessment of underlying musculature, as well as cartilaginous and osseous structures. Visceral organs and tissues, including gills, intestine, kidney, spleen, liver, heart, and brain, were fixed for 24 h in 10% neutral buffered formalin (NBF) (Merck, Gillingham, UK) and stored afterward in 70% ethanol for histopathology examinations. Tissues were embedded in paraffin wax using the PELORIS II Premium Tissue Processing System (Leica Microsystems, Gillingham, UK). Embedded blocks were then sectioned at 3−4 μm thickness with a Shandon Finesse rotary microtome (Fisherscientific, Gillingham, UK) and stained with haematoxylin and eosin (H&E) following standard protocols. Tissue sections were examined with a Nikon Ni-E brightfield microscope (ThermoFisherScientific, Edison Road, UK) with images captured using Nis-Elements software (Nikon, Gillingham, UK). Following initial examinations, a selection of samples was re-cut for Giemsa and Gram stains for the elucidation of protists and bacteria in tissues. Staining methods were performed following standard protocols [73].

### 4.3. Bacteriology and Virology Examinations

Swabs taken from the jaw (lesions) and head kidney were plated either onto tryptic soy agar (TSA) [74], Mueller-Hinton agar (MHA) [75], and tryptone yeast extract salts (TYES) [76] agar. Plates were then incubated at 15 °C and observed daily. 

During the sampling, dissected tissues were placed in 1:10 transport media (‘Glasgow’s MEM, supplemented with 10% foetal bovine serum, 200 IU mL^−1^ penicillin, 200 μg mL^−1^ streptomycin, and 2 mM L-glutamine (Gibco, Paisley, UK)) and kept for 24 h at 4 °C. From each sampled fish a jaw section including dermis and muscle tissues and a pool of head kidney, spleen, heart, and brain were homogenized 1:10 (weight/volume) in transport media. Homogenates were clarified by centrifugation for 10 min at 2500× *g* and inoculated at a dilution of 10^−2^ and 10^−3^ onto the following fish cell lines: chinook salmon embryo (CHSE-214) (ATCC^®^: CRL-2872™) [77]; Atlantic salmon head kidney (TO) [78]; epithelioma papulosum cyprini (EPC) (ATCC^®^: CRL-2872™) [79]; and bluegill Lepomis macrochirus fry (BF-2) (ATCC^®^: CCL-91™) [79]. Inoculated cells were maintained at 15 °C for 14 days with regular observation for the development of CPE, followed by a blind passage and incubation for a further 14 days.

In parallel, tissue homogenates were pre-treated with infectious pancreatic necrosis virus (IPNV) neutralizing antisera (Polyclonal goat anti-Serotype Sp IPNV serum; Harlan Sera-lab, Gillingham, UK) and inoculated onto cells lines as described above. Samples developing CPEs were subjected to an IPNV ag ELISA test (TestLine Clinical Diagnostics, Czech Republic) according to the manufacturer’s protocol.

### 4.4. PCR Detection of Viral Pathogens

Viral nucleic acid was extracted from 200 µL of the supernatant of RTG-2 cells showing CPEs using a Maxwell^®^ RSC viral total nucleic acid purification kit in a Maxwell^®^ RSC instrument (Promega, Southampton, UK) following the manufacturer’s instructions. Reverse transcription (RT) was performed at 37 °C for 1 h in a total 20 µL volume consisting of 200 U of M-MLV RT, M-MLV RT 5x reaction buffer (250 mM Tris-HCl, pH 8.3; 375 mM KCl; 15 mM MgCl2; 50 mM DTT), 1 mM dNTP mix, 500 ng of random primers, 25 units RNasin^®^ Ribonuclease Inhibitor (Promega, Hampshire, UK) and 4 µL of RNA. 

A set of generic PCRs for the molecular diagnostic of viruses in the genera *Novirhabdovirus*, *Aquabirnavirus*, *Iridovirus*, and *Vesiculovirus* was conducted as described before [80]. PCRs were performed in a 50 µL volume consisting of 10x green GoTaq^®^ Flexi buffer, 2.5 mM MgCl_2_, 1 mM dNTP mix, 1.25 units of GoTaq^®^ G2 Hot Start Polymerase (Promega, Hampshire, UK), 50 pmol of each forward and reverse primers, and 2.5 µL of either DNA or cDNA. PCR amplification included a denaturing step of 5 min at 95 °C followed by 35 cycles of 1 min at 95 °C, 1 min at 55 °C, 1 min at 76 °C, followed by a final extension step of 10 min at 72 °C in a Mastercycler nexus X2 (Eppendorf, Stevenage, UK). PCR products were analysed in a 2% (weight/volume) agarose/TAE gel (40 mM Tris-acetate, pH 7.2, 1 mM EDTA, 1.0 μg mL^−1^ ethidium bromide) at 120 V, 400 mA for 30 min and visualised under UV light using the Gel Dox Imager XR+ (Bio-Rad, Hemel Hempstead, UK).

### 4.5. Illumina MiSeq Next-Generation Sequencing

Cell culture displaying CPEs were selected for high-throughput sequencing. Total DNA and RNA were extracted from the culture using either a Maxwell^®^ RSC Tissue DNA Kit or a Maxwell^®^ RSC simplyRNA Tissue Kit in a Maxwell^®^ RSC instrument (Promega, Hampshire, UK) following the manufacturer’s instructions. In parallel, clarified supernatants were pelleted by ultracentrifugation at 100,000× *g* for 1 h at 4 °C and re-suspended in 200 μL of cell culture medium. Viral nucleic acid was then extracted and cDNA (reverse transcription) was obtained as described above. Double-stranded cDNA was then generated with random primers using Sequenase V2.0 DNA Polymerase (Affymetrix, ThermoFisherScientific, Edison Road, UK). Libraries were prepared with the Nextera XT DNA Library Preparation Kit (Illumina, Great Abington, UK) and sequenced with Miseq v2 Reagent Kit 2× 300 bp paired-end protocol (Illumina, Great Abington, UK) following the manufacturer’s recommended procedures in an Illumina MiSeq platform).

Analysis of the raw data, reads alignment, and alignment of the consensus sequences against the following reference sequences: VHSV 23–75 (GenBank acc. no. FN665788), IPNV segment A (GenBank acc. no. L40583.1), EHNV (GenBank acc. No MT510743.1), SAV2 or slepping disease virus (GenBank acc. NC_003433.1), and *T.bryosalmonae* 18S ribosomal RNA gene (GenBank acc. no. EU570235.1) were done using CLC Genomics Workbench v4.9 (Qiagen, Manchester, UK) and the bioinformatic pipeline described in [81].

In addition, the most abundant reads (top 25) were aligned against the NCBI database using a nucleotide blast search (NCBI nBlast database accessed in December 2020). The threshold for filtering blast results was 80% identity.

### 4.6. Taqman qPCR Analysis

For each sampled fish, both jaw sections (Appendix A) and head kidney (to study possible systemic responses) were fixed in RNA-later for the gene expression analysis of 17 immune-related genes in rainbow trout. Those genes were: *interleukin 1b* (*il1b*), *interleukin 4/13a* (*il4/13a*), *interleukin 10* (*il10*), com*plement c3* (*c3*), *lysozyme II* (*lyz2*), the transcription factors *T-bet* (*tbx21*) and *GATA-binding protein 3* (*gata3*), *immunoglobulin M heavy chain* (*IgM*), *type I interferon* 3 (*ifn3*), *viperin* (*vig1*), *cluster of differentiation 8* (*cd8a*), *Toll-like receptor 3* (*tlr3*), *transforming growth factor beta* (*tgf-b*), *heat shock* 70 *kDa protein 12A-like* (*hspa12a*), as well as specific cellular markers reported to be up-regulated during proliferative kidney disease: the *interleukin 11* (*il11*) and *cathelicidin 1* (*cath1*) [82]. Taqman qPCR assays were described before except for *il11*, *cath1*, and *hsp12a* assays which were de novo designed (Table 2 and Table 3). 

A Taqman qPCR assay for the detection of the *T. bryosalmonae 60S ribosomal protein L18* (*rpl18*) was also designed. The gene expression of rainbow trout *beta*-*actin* (*b-actin*) (AJ438158.1) was used as a reference gene for normalization purposes.

Taqman qPCR assays were conducted in duplicate in a 20 µL master mix containing 2 µL of the cDNA, 500 nM of each primer, and 250 nM of probe labelled with 6-FAM 5′ and 3′ BHQ1, and 10 µL of the 2x Taqman^®^ Universal Master Mix II with UNG (Applied Biosystem, Southampton, UK). Real-time PCR reactions were performed on a StepOne Real-Time PCR apparatus with V2.3 software (Applied Biosystem, Southampton, UK) with 40 cycles of PCR amplification at an annealing temperature of 60 °C and fluorescence detection as recommended by the manufacturer. Molecular grade water was used as a negative control for each master mix.

### 4.7. Statistical Analysis

Serial tenfold dilutions of a cDNA sample were used to generate standard curves to determine each primer set efficiency, giving slope values close to −3.2. The inter-run calibrated normalized relative quantities (CNRQ) in the gene expression, equivalent to the fold change method (2−^∆∆Ct^) [85] were calculated using qbase+ version 3.2 software (Biogazelle, UK) [86]. An analysis of the variance was assayed in an ANOVA test to determine significant differences in the gene expression (*p* value < 0.05) between groups and control fish. The normalised gene expression patterns were visualised through a heatmap plot generated in R, version 3.6.1 [87]. 

## Figures and Tables

**Figure 1 pathogens-10-00542-f001:**
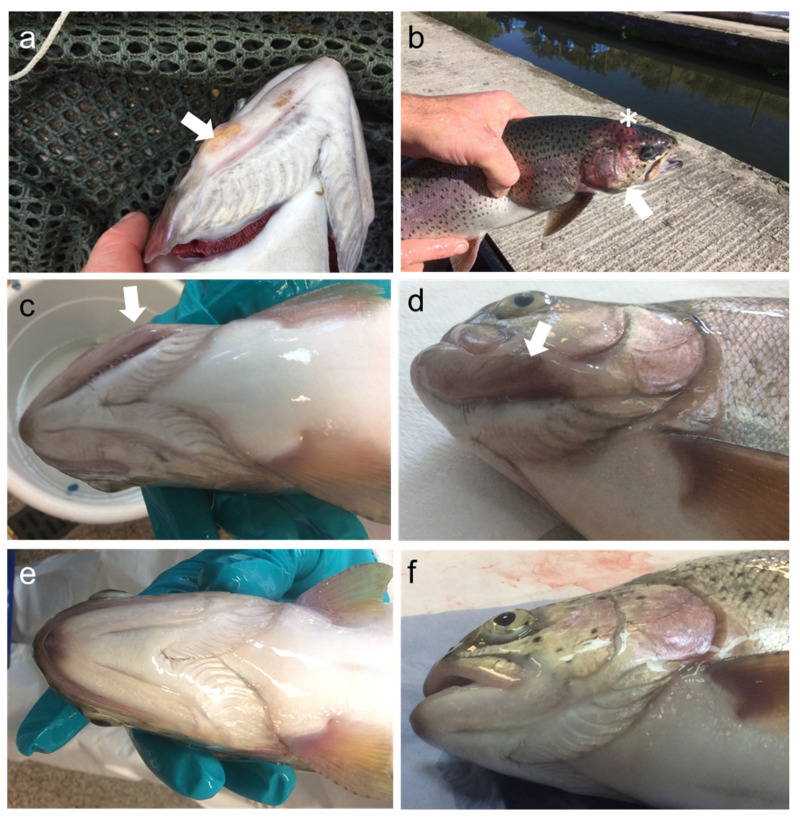
Gross clinical signs of cranial mandibular fibrosis syndrome. Rainbow trout sampled in 2018 showing: (**a**) discoloured nodular swelling on the jaw (arrow); (**b**) erythematous change to the “cheek” (asterisk) with marked swelling particularly to cranial tissue posterior to the eye and involving the operculum (arrow). Fish sampled in 2019 showing: (**c**,**d**) unilateral erythema and swelling in the region of the jaw; (**e**,**f**) unaffected fish sampled in the same farm, 2019.

**Figure 2 pathogens-10-00542-f002:**
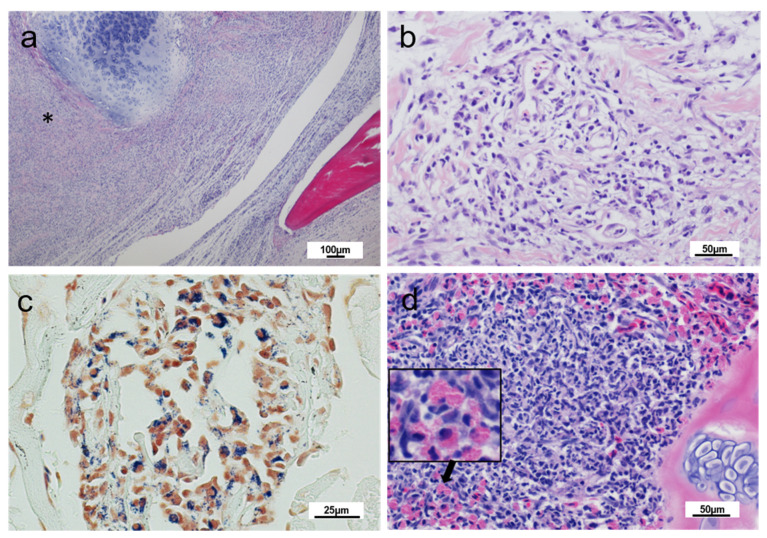
Histopathology of rainbow trout affected with cranial mandibular fibrosis syndrome; fish sampled in 2018. (**a**) The jaw section showing a massive inflammatory infiltrate and fibrosis between cartilage and osseous elements (*); (**b**) Detail of the dermis of affected fish showing inflammatory infiltration; (**c**) Section of a heart showing Gram-positive bacteria within a foci of infiltration of phagocytic-cell type cells; (**d**) Gill section showing an increased number of eosinophilic granular cells (insert); (**a**,**b**,**d**) Haematoxylin and eosin stain, (**c**) Gram stain.

**Figure 3 pathogens-10-00542-f003:**
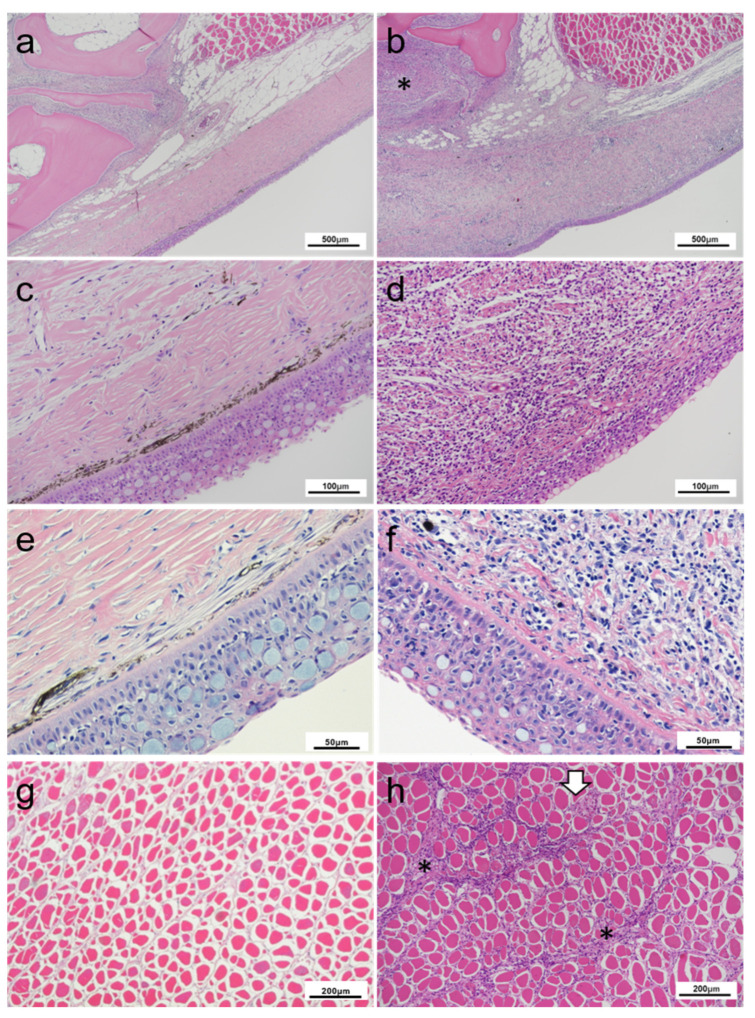
Histopathology of rainbow trout affected with cranial mandibular fibrosis syndrome (CMF) from fish sampled in 2019. (**a**,**c**,**e**) Skin section of clinically healthy fish; Dermal layers are undisrupted; (**b**,**d**,**f**) Skin sections through mandibular region of affected fish; (**b**) Extensive inflammatory infiltration and cellularity disrupting the dermal layer for increased basophilia of the region, extending into subcutaneous tissues (*); (**d**,**f**) Detail of inflammatory cell infiltration and connective fiber necrosis for disruption of dermal structures; (**g**) Underlay musculature of healthy fish; (**h**) Inflammatory infiltration of the underlying musculature within the perimysium (*). Muscle fibers remain largely intact besides a foci inflammation (white arrow). (**a**–**h**) Haematoxylin and eosin stain.

**Figure 4 pathogens-10-00542-f004:**
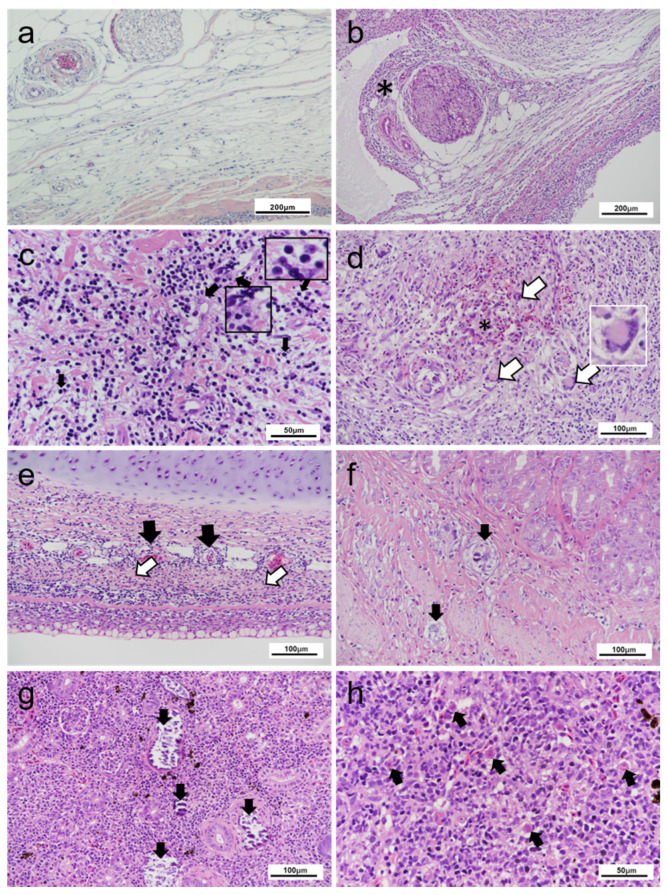
Pathological changes observed in rainbow trout displaying cranial mandibular fibrosis (CMF) syndrome. (**a**) Skin section of clinically healthy fish showing neural fibres; (**b**) Inflammatory infiltration (*) of the epineurium and neural tissue of CMF affected fish; (**c**) Necrosis with karyorrhexis debris alongside mixed inflammatory infiltration. Histiocytes, lymphocytes (black arrows and inserts) visible within disrupted dermal tissue; (**d**) High magnification section of less-typical presentation of CMF. Dermal tissue with focal haemorrhage (*) adjacent to multinucleated giant cells (arrows) in the region of the branchiostegal rays (‘folds’ region) of jaw section. Top right inset: detail of a multinucleated giant cell with nuclei arranged at one pole; (**e**) Perivascular infiltration by immunological cells. Dermatitis is accompanied by necrosis in the dermis (underlying cartilage is unaffected). Areas of necrosis (white arrows), inflammatory cell infiltration around the vascular structures (black arrows); (**f**) Calcinosis (arrows) of lamina propria and smooth muscle layers of the gastrointestinal tract; (**g**) Nephrocalcinosis (arrows) seen within the renal tissue; (**h**) PKD-infected head kidney with multifocal *Tetracapsuloides bryosalmonae* infection (arrows). (**a**–**h**) Haematoxylin and eosin stain.

**Figure 5 pathogens-10-00542-f005:**
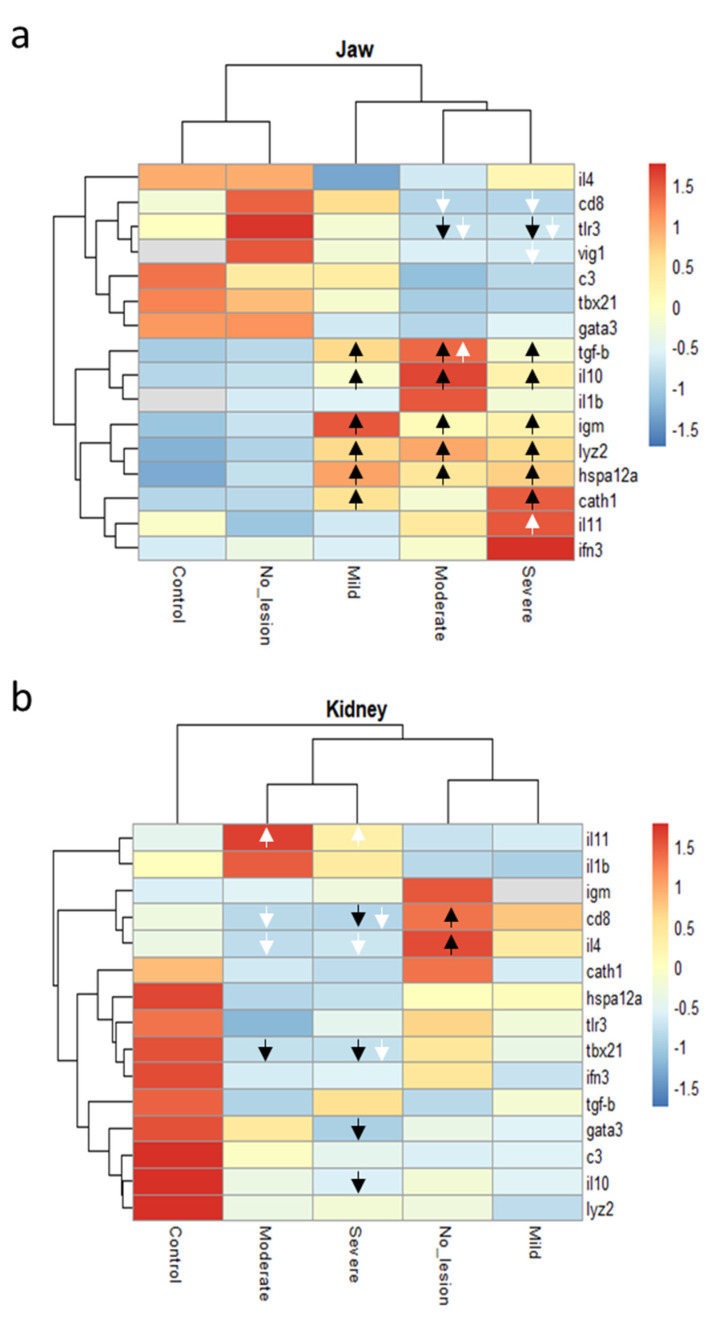
Heatmap dendrograms showing clustering of genes (mean values of normalized relative quantities) either in the jaw (**a**) or kidney (**b**) sections of rainbow trout displaying mild (4 fish), moderate (4 fish), or severe (8 fish) clinical signs of cranial mandibular fibrosis syndrome. No_lesion group (7 fish) refers to fish with no apparent lesions sampled in the affected site. The same fish stock was sampled from a nearby unaffected farm and used as negative controls (7 fish). Arrows show the direction (up or down-regulated) of the genes that were differentially expressed (*p* < 0.05) either from the control group (black arrows) or the no-lesion group (white arrows).

**Table 1 pathogens-10-00542-t001:** Overview of histopathological changes observed in rainbow trout affected with cranial mandibular fibrosis (CMF) syndrome sampled in 2018 and 2019. *n* = number of samples with CMF. Dermatitis refers to inflammation of the dermal tissues. NAD: no abnormalities detected. PKD: proliferative kidney disease, observed presence of *Tetracapsuloides bryosalmonae* infection.

	2018 (Spring)	2018 (Autumn)	2019 (Spring)
*n*	4	10	16
Jaw	Dermatitis, fibrosis (4/4)	Dermatitis, fibrosis (10/10)	Dermatitis, fibrosis (16/16) Epineurium infiltration and vasculitis (3/16)
Gill	Epithelial hyperplasia and eosinophil granule cells (2/4) *Saprolegnia* (1/4)	NAD	Epithelial hyperplasia (8/16)
Liver	Focal inflammatory cell infiltration (1/4)	Focal inflammatory cell infiltration (5/10) PKD (5/10)	Focal inflammatory cell infiltration (1/16) PKD (1/16)
Spleen	NAD	Focal inflammatory cell infiltration (3/10) PKD (5/10)	NAD
Kidney	Focal inflammatory cell infiltration (1/4)	Multifocal inflammatory cell infiltration (1/10) PKD (5/10) Nephrocalcinosis (1/10) Neutrophils (1/10)	Focal inflammatory cell infiltration (2/16) PKD (1/16) Nephrocalcinosis (1/16)
Heart	Mild myocarditis (2/4)	Myocarditis (1/10) Gram-positive bacteria infection (1/10)	NAD
Stomach	NAD	Lymphocytic infiltration (1/4) Calcinosis (1/4)	Calcinosis (2/16)

**Table 2 pathogens-10-00542-t002:** Summary of the rainbow trout genes analysed in this study.

Gene	Gene Involvement	GenBank acc. no	Reference
*vig1*	IFN inducible protein, antiviral	AF076620.1	[83]
*tlr3*	Innate sensing of viral dsRNA	NM_001124578.1	[69]
*tgf-b*	Cell growth and proliferation	X99303	[84]
*tbx21*	Promotes Th1 polarization as a response of intracellular pathogens	FM863825	[84]
*igm*	B cells antigen recognition	S63348.1	[83]
*lyz2*	Antibacterial protein	X59491	[84]
*il11*	Platelet and cell proliferation	AJ867256.1	This study
*il4/13a*	Promotes Th2 polarization as response of extracellular parasites)	AB574337	[84]
*il1b*	Pro-inflammatory cytokine, promotes Th17 polarization	NM_001124347.2	[83]
*ifn3*	Antival activity	NM_001160502.1	[69]
*hspa12a*	Cellular stress and toxic chemicals	XM_021580438.1	This study
*gata3*	Promotes Th17 polarization	FM863826	[84]
*cath1*	Antimicrobial peptide, pro-inflammatory	AY382478.1	This study
*c3*	Complement system	AF271080	[84]
*il10*	Immunoregulatory	AB118099	[84]
*cd8*	Marker of cytotoxic T lymphocytes	AF178054	[84]

**Table 3 pathogens-10-00542-t003:** Primers and probes used for Taqman qPCR assays for the detection of *Tetracapsuloides bryosalmonae 60S ribosomal protein L18* (*rpl18*) (GenBank acc. no. FR852769.1) and the rainbow trout *interleukin 11* (IL11) (AJ867256.1), *cathelicidin-1* (*cath1*) (AY382478.1), *heat shock protein 70 kDa protein 12A-like* (*hspa12a*), and *beta-actin* (*b-actin*) (AJ438158.1) mRNA.

Gene	Forward 5′-3′3′	Reverse 5′-3′3′	Probe (Fam) 5′-′3′ (BHQ-1)
*rpl18*	CACTGTTATTGCAGGGCTGTAGA	TGGAGCAGCACCAAAATACCT	AGGCCAGGGTTGC
*il11*	TGAGTGTCTGTCTATCGTCACTATCAGT	AGGGCGAACAATCCAATAAAGA	TTTACGGAACAAAAAGTTTGGAG
*cath1*	CTTTGCCTCAGCTGCTTCCT	TGGAGCTGGTTCAGAATTGGA	AGAGCAGGCTTTCC
*hspa12a*	AGCGGACGCCCCAAA	TCCTCAGGGTAGAAGCTCTTGGT	TGGAGGTTGAATACAAAG
*b-actin*	GAAATCGCCGCACTGGTT	CGGCGAATCCGGCTTT	TTGACAACGGATCCGGT

## Data Availability

Data is contained within the article or Appendix A.

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
