# Peer review of "Cranial Mandibular Fibrosis Syndrome in Adult Farmed Rainbow Trout Oncorhynchus mykiss"

_pathogens, 2021, doi:10.3390/pathogens10050542_

Round 1

Reviewer 1 Report

In the manuscript, several times it is written "inflammation", "inflammatory infiltrate", "inflammatory response", "inflammatory lesions", "inflammatory reaction", "inflammatory infiltration", "inflammatory infiltrative cell types", "extensive inflammatory infiltration", "perivascular infiltration", "inflammatory aggregates" . Most of the times there are no details of the inflammation and of the cells that are present, including in the legends of Figures. Only sometimes there is a reference to cells, namely macrophages, histiocytes, lymphocytes and neutrophiles.

In Figures, details of the cells should be present, visible and identified.

Line 153 - What is "Cardiomyositis"? What are the features of "Cardiomyositis"? Which are the lesions that were observed?

How was fibrosis and collagen identified? In line 170, "collagen" is mentioned. Any histochemical exam was performed? Any special stain was applied in tissues sections?

Line 180-182 - Authors refer that "multinucleate giant cells" do not appear to be a consistent feature of pathology. However, their presence should be discussed by Authors. In Line 195, Authors also refer the presence of "multinucleated giant cells".

Line 192 - Which are the features of "Dermatitis"? The description of the lesions should be detailed.

Line 197 - The word "lympohcytes" must be corrected.

Lines 211-212 - What do you think is the cause of "Calcinosis" and Nephrocalcinosis"? It is important to discuss these lesions.

Lines 390-391 - Why do Authors wrote "Calcinosis is a chronic inflammatory condition"? What does it mean exactly? Calcinosis is mineralization  of tissue in which calcium is deposited. In the lesions observed by Authors, inflammatory cells, were they present? 

Line 418 - What is "inflammatory fibrosis"? 

Line 433 and Table 3 - It is written "inflammatory cytokines". Can cytokines be "no inflammatory"?

Lines 461-462 - It is written "pro-inflammatory and pro-fibrosis cytokines". What is the difference between pro-inflammatory and pro-fibrosis cytokines?

Author Response

Dear Reviewer no. 1,

Thanks for your time in peer-reviewing this manuscript and for your notes. We have updated the text accordingly and updated figures and legends. We hope the clarity of the text has improved.

In the manuscript, several times it is written "inflammation", "inflammatory infiltrate", "inflammatory response", "inflammatory lesions", "inflammatory reaction", "inflammatory infiltration", "inflammatory infiltrative cell types", "extensive inflammatory infiltration", "perivascular infiltration", "inflammatory aggregates" . Most of the times there are no details of the inflammation and of the cells that are present, including in the legends of Figures. Only sometimes there is a reference to cells, namely macrophages, histiocytes, lymphocytes and neutrophiles.

We have attempted to resolve any confusion on the nature of the inflammation present within various tissues. These changes include; replacing description of ‘inflammation’ in clinical/gross features with more specific terminology (swelling and erythema), and enhancing the description of immune cell infiltrations in the description of histopathology. The site of the CMF (head/jaw) pathology contained a mixed immune cell presence dominated by histiocyte + lymphocyte cell types, and the text has been modified to emphasise this fact. Some of the more incidental/less consistent findings however featured different cell-type inflammatory infiltrations, and the text has been altered to reflect this too.

In Figures, details of the cells should be present, visible and identified.

We have included in figure 4 detail of the immune cells observed. 

Line 153 - What is "Cardiomyositis"? What are the features of "Cardiomyositis"? Which are the lesions that were observed?

Cardiomyositis refers to the inflammation of the myocardium. We have changed the term for clarity. The pathology is shown in Figure 1b  and described on the legend as “Section of the heart showing the presence of large numbers of Gram-positive bacteria within a foci of inflammation”.

How was fibrosis and collagen identified? In line 170, "collagen" is mentioned. Any histochemical exam was performed? Any special stain was applied in tissues sections?

This is a typographical error. Collagen should read here ‘connective tissue’. This has been remedied in the text.

Use of Giemsa and gram stains were applied in this study. We also conducted Masson's trichrome staining and use of van geison staining to visualise elastin, however, these stains are not of publication quality. Although the fibrotic change to tissue appears to be collagenous in nature based on Masson's trichrome staining, without high quality illustrations of this assertion it was not felt that this aspect could be explored in any great depth. 

Line 180-182 - Authors refer that "multinucleate giant cells" do not appear to be a consistent feature of pathology. However, their presence should be discussed by Authors. In Line 195, Authors also refer the presence of "multinucleated giant cells".

We have corrected the term to “multinucleated giant cells” and added an insert in figure 4 to show it in detail.  We have added a  paragraph to discuss its presence.

Line 192 - Which are the features of "Dermatitis"? The description of the lesions should be detailed.

The section previously on line 192 was intended as a summary of the above histopathology findings, all of which detail the dermatitis (in extent, cell type, and other features, including necrosis, of dermal tissue layers). We apologise that is was not clear, and have expanded on this section to remove the attempt at a ‘summary’, and instead ensure the the pertinent findings are clear from the text as a whole. Hopefully this clarifies the ‘typical’ findings of this condition, based on various sampling visits. The overall dermatitis (dermal inflammation) is not characterised beyond being described as chronic (based on the immunological cell types observed and protracted nature of the clinical presentation) and in its extend (which is diffuse). It did not seem appropriate to characterise the dermatitis further (other than through the provision of a detailed description) as the presentation was not considered sufficiently similar to other, previously described, discrete dermal pathologies in salmonids.

Line 197 - The word "lympohcytes" must be corrected.

Corrected

Lines 211-212 - What do you think is the cause of "Calcinosis" and Nephrocalcinosis"? It is important to discuss these lesions.

We have expanded the relevant discussion paragraph.  The main drivers of calcinosis are water  quality and diet, in particular high levels of carbon dioxide in water, diet deficiencies causing low magnesium and toxicity to selenium and arsenic. Recently it has been noted in fish infected with carnobacterium (added in line 478). 

Lines 390-391 - Why do Authors wrote "Calcinosis is a chronic inflammatory condition"? What does it mean exactly? Calcinosis is mineralization  of tissue in which calcium is deposited. In the lesions observed by Authors, inflammatory cells, were they present? 

Thanks for driving our attention to this paragraph. We have deleted “chronic inflammation” from the definition of calcinosis and added references to support the paragraph's statements. We did not observed a direct inflammatory response associated with calcinosis.

Line 418 - What is "inflammatory fibrosis"? 

Fibrosis refers to the  pathology derived from a chronic inflammation. For clarity, we have changed “inflammatory fibrosis” for “chronic inflammation.”

Line 433 and Table 3 - It is written "inflammatory cytokines". Can cytokines be "no inflammatory"?

Some cytokines trigger inflammatory processes while others counterbalance towards torelance (anti-inflammatory). We have modified the term “inflammatory”  for “pro-inflammatory” for clarity. (Literature of interest: Opal SM, DePalo VA. Anti-inflammatory cytokines. Chest. 2000 Apr;117(4):1162-72. doi: 10.1378/chest.117.4.1162. PMID: 10767254.)

Lines 461-462 - It is written "pro-inflammatory and pro-fibrosis cytokines". What is the difference between pro-inflammatory and pro-fibrosis cytokines?

The difference are the networks (pathways) in which those cytokines are involved.  Not all pro-inflammatory cytokines will induce fibrosis. However, key pro-inflammatory cytokines involved  in the TGF-B  cand T-reg polarization (as tgf-b, il10, il11) strongly induces fibrosis  (references in the discussion, lines 440-444).

Reviewer 2 Report

The absence of information in unnotified diseases to the OIE implies the lack of valid information in the everyday life aquaculture diseases, so describing fish diseases presumably provoked by pathogens de novo seems to be of relevance. After reading the manuscript, my recomendation is to accept it after minor revision based on completing the information given and the bibliographic background in the discussion. 

Table 1: NA abbreviation has not been described.

Figure 2: Since the sections shown are from pathogenic diseases and author have described that fish with no pathology were also found, stained section from fish with no lesions (line 117 fig. 1e,f) should be included as they did in Figure 3.

Figure 3: Images a and b have a different scale bar. It would be useful for the readers to include also comparative regions of the ones corresponding to disease in the same scale. Moreover, some scale bars are too small to be read, they all should have the same letter size. The inclusion of images from non-affected region for comparative analysis should be included in all figures.

Line 207: Microscopic observation of blood smears did not show any abnormalities should include “data not shown” if authors are not including the information in any format.

Figure 5: Again, an image from control RTG2 monolayer should be included.

Lines 429-432: Authors mention that the up-regulation of antimicrobial peptides-related genes correlates with the high number and leucocytes infiltration. Although the relationship with the cell infiltration during an inflammatory process into the tissues is mentioned, authors omit the background (bibliography) that could correlate antimicrobial peptides with those processes in fish, so this correlation in studied fish appears not to be justified in this work in the way in which discussion is written. Whether the mechanisms by which antimicrobial peptides act in each fish species is still not completely defined, authors should correlate all those processes properly.

Discussion: In general, the discussion point should be better supported by bibliography.

Authors should include a justification of why they chose those immune-related genes. Is it only according to the fish symptoms?

Author Response

The absence of information in unnotified diseases to the OIE implies the lack of valid information in the everyday life aquaculture diseases, so describing fish diseases presumably provoked by pathogens de novo seems to be of relevance. After reading the manuscript, my recomendation is to accept it after minor revision based on completing the information given and the bibliographic background in the discussion. 

Dear Reviewer no. 2,

Thank you for your time in peer-reviewing this manuscript and your notes. We have updated the text accordingly, updated figures and legends, and done extensive modifications in the discussion. We much hope that the clarity and quality of the text have improved.

Table 1: NA abbreviation has not been described.

This was a typographical error, corrected for NAD.

Figure 2: Since the sections shown are from pathogenic diseases and author have described that fish with no pathology were also found, stained section from fish with no lesions (line 117 fig. 1e,f) should be included as they did in Figure 3.

Lamentably, in 2018 only  affected specimens were received either from a private consultant or by the FHI for histopathology assessment. In 2019, however, we collected both affected and non-affected fish.

Figure 2 shows pathological changes observed in fish from spring 2018. Fish with no lesions were no received in 2018 hence the lack of controls in the figure.

Figure 3 shows pathology observed in 2019. In this occasion, we sampled both non-affected and affected fish, as shown in the images.

We have amended the text in results sections 2.1 and 2.2 and M&M (lines 507-520)  for  clarity.

Figure 3: Images a and b have a different scale bar. It would be useful for the readers to include also comparative regions of the ones corresponding to disease in the same scale. Moreover, some scale bars are too small to be read, they all should have the same letter size. The inclusion of images from non-affected region for comparative analysis should be included in all figures.

We sincerely apologise that the numbers on the scale were too small to read. We have amended them in Figures 2-4.

Figure 3 has been amended to show images of affected and healthy fish at the same scale. 

The inclusion of images from non-affected region for comparative analysis should be included in all figures.

Images from control fish at different magnifications are now included in figure 3 for comparison with figures 2 and 3. In  Figure 4, a control image of neural fibres in the jaw section has been added. The figures showing calcinosis and Tetracapsuloides bryosalmonae are labelled to show the calcium deposits and the parasitic cells.  

Line 207: Microscopic observation of blood smears did not show any abnormalities should include “data not shown” if authors are not including the information in any format.

Due to the lack of abnormal cells, photos were not taken. “Data not shown” added in the text.

Figure 5: Again, an image from control RTG2 monolayer should be included.

Lamentably we have deleted this image as requested by reviewer no. 3.

Lines 429-432: Authors mention that the up-regulation of antimicrobial peptides-related genes correlates with the high number and leucocytes infiltration. Although the relationship with the cell infiltration during an inflammatory process into the tissues is mentioned, authors omit the background (bibliography) that could correlate antimicrobial peptides with those processes in fish, so this correlation in studied fish appears not to be justified in this work in the way in which discussion is written. Whether the mechanisms by which antimicrobial peptides act in each fish species is still not completely defined, authors should correlate all those processes properly.

We have added a bibliography background paragraph on the role of AMPs as immunomodulators in fish (lines 465-472).

Discussion: In general, the discussion point should be better supported by bibliography.

We have amended the discussion to support the statements with bibliography references.

Authors should include a justification of why they chose those immune-related genes. Is it only according to the fish symptoms?

Yes. In this study, histology was done before the molecular work. Therefore, we knew that the fish showed a sterile chronic inflammation (we could not identify a pathogen in the lesions).  The 17 genes included in this study are key genes involved in immune responses in rainbow trout. We used these genes as biomarkers to reveal which pathways were up-regulated, which might inform on the cause of the inflammation.  The genes that were studied included pro-inflammatory cytokines (IL-1β) and cytokines associated with adaptive immunity (IFN-3, IL-4/13, IL-10, TGF-β).  These cytokines and transcription factors (tbx21 and gata3) also can inform on the T-cell polarization  and nature of a pathogen (typically Th1-intracellular pathogens; Th2- ectopic parasites; Th17- extracellular pathogens; Treg- tolerance/immunoregulation). Also we included antimicrobial peptides, complement, and lysozyme in our study. We hope to develop cost-effective microarrays to study a larger set of host genes in disease investigations in the future. We have included a paragraph in the discussion to clarify why these genes were selected as biomarkers (lines 637-342).

Reviewer 3 Report

The manuscript «Cranial maxillary fibrosis syndrome in adult farmed rainbow trout Oncorhynchus mykiss» gives some interesting aspects into the pathology of this special syndrome. The manuscript also aims to rule out possible causes for initiation of the lesions. It is a well written manuscript, however, there are some major concerns which have to be addressed before publication.

General comments

The authors invested a lot of work in investigating possible causes for the syndrome. In that respect it is quite sad, that authors limited their investigations by e.g. investigating only cell culture supernatants for NGS instead of on affected jaw tissue. By NGS, authors only searched for special viruses instead of for unknown agents. In addition, indications for Renibacterium salmoninarum infections (gram positive bacteria intracellular in macrophages in the heart are very much indicative for Renibacterium salmoninarum) were not followed up.

In addition, this report is pretty much based on the pathological investigations. However, in the description of the histopathology there were several inconsistencies and unclear/missing/wrong descriptions.

Some special comments:

Abstract

Correlation of the syndrome to nephroclacinosis and PKD is not clear and could be deleted from the abstract

Results

Page 3, line 119: please use "inapetence" instead of innapetenced

Page 4: Histopathology: which inflammatory cells were found? Please be more specific.

Eosinophilia in the gills? Please be more specific. Which cells show hypereosinophilia?

Fouling of gills? Do authors mean autolysis?

What is perivascular eosinophilia?

Epicarditis is composed of which cells?

Table 1: how many animals were examined? How many animals showed lesions? Please give numbers? Again, the terms in the table need to be explained, in the table or in the text, e.g. dermatitis – modifier, severity, etc.. Eosinophilia is not a patho term. What do authors mean? Lamina propria of which organ? Intestine?

Authors describe neutropilic infiltration in association with PKD. Neutrophils is not the prominent inflammatory cell with PKD. What did the authors see?

The cardiomyositis with the gram positive bacteria resembles very much BKD, an infection with Renibacterium salmoninarum. This should be clarified, as BKD is a chronic disease with a long incubation period of several months inducing granulomatous inflammation in several organs. This infection should be ruled out.

Figure 2a: it would be worth to show a higher magnification as well. Please be more specific which inflammatory cells were involved.

Figure 3: same, please specify which immune cells were involved

Figure 4 c-f can be deleted

Microbiology findings: first sentence belong to Materials and Methods

Figure 5 can be deleted

Page 8, line 258: RNA instead of ARN

Host response: Authors examined immune gene expression in the kidney: was this correlated to infection with T. bryosalmonae? If fish were additionally infected with this parasite, immune gene expression is most probably related to the parasite infection instead of to the CMF syndrome.

Figure 7 can be deleted

Discussion

Page 13, line 390: what is a "stomach linen"? Serosa? Mucosa?

Page 13, line 404: how can authors justify this statement? To confirm the absence of pathogen, a more thoroughly examination would be needed.

Line 406: there are also infectious agents with an incubation time of several month, e.g. Renibacterium salmoninarum

The upregulation of the immune genes in the skin and kidney is not surprising as we see a mononuclear infiltration in the skin and an infection with T. bryosalmonae in the kidney. However, what is the interpretation of the results? Are there any indications for possible causes?

Page 14, line 463: in my opinion, an association between the "new" syndrome and PKD is very questionable. More probably the PKD is associated with the sewage plant inlet without any association to the jaw changes.

Materials and Methods

Page 15, line 503: Giemsa and gram stains are very superficial to rule out protists and bacteria. Especially, if the authors find Gram positive bacteria and do not go further!

Lines 521-525: why only go further in detail with IPN? Was there any suspicion for IPN?

Virology and PCR: on which criteria were the viruses selected for further investigation? Did authors really expect these viruses to be involved in the jaw lesion? And if yes, why?

NGS: it is pretty much sad that authors limited the NGS examination on the cell culture supernatants of CPE positive cell cultures. Why limiting the examination on viruses which grow on the cell lines?

Author Response

The authors invested a lot of work in investigating possible causes for the syndrome. In that respect it is quite sad, that authors limited their investigations by e.g. investigating only cell culture supernatants for NGS instead of on affected jaw tissue.

Dear Reviewer, thank you for your time in peer-reviewing this manuscript and your notes. We did not see any pathogen in the jaw lesions that justified the cost of NGS in the tissue. To sequence  directly from the tissue, we would have needed a deeper sequencing platform as HiSeq instead of MiSeq.

 By NGS, authors only searched for special viruses instead of for unknown agents.

We mapped the consensus sequences against viruses pathogenic to rainbow trout, but also, we blasted the reads against everything available on the NCBI (in M&M). All the reads matched the host genome. If there were reads from an unknown pathogen that is not yet available on the NCBI we could have overlooked it. The sequencing data can be revisited in the future as sequences of novel pathogens become available. 

In addition, indications for Renibacterium salmoninarum infections (gram positive bacteria intracellular in macrophages in the heart are very much indicative for Renibacterium salmoninarum) were not followed up.

We lament that this sample was not sequenced. The bacteria infection was seen in a single fish in 2018. We did not see bacteria infection in the rest of the fish sampled in 2018 or 2019 showing severe pathology in the jaw. Similarly, we could not isolate any bacteria in agar plates from those fish, although we acknowledge that Renibacterium salmoninarum is a particular fastidious bacterium to grow in vitro. Typical clinical signs of bacterial kidney disease was never observed on this farm in any of the visits by the FHI and neither reported by the resident vet. 

In addition, this report is pretty much based on the pathological investigations. However, in the description of the histopathology there were several inconsistencies and unclear/missing/wrong descriptions.

We have amended the text to resolve any inconsistencies or missing descriptions within the pathological report.

Some special comments:

Abstract

Correlation of the syndrome to nephroclacinosis and PKD is not clear and could be deleted from the abstract

We have modified the abstract to describe which other pathologies were found and that we could not establish a correlation of these pathologies with the syndrome.

Results

Page 3, line 119: please use "inapetence" instead of innapetenced.

Thanks, corrected.

Page 4: Histopathology: which inflammatory cells were found? Please be more specific.

The text has been altered to emphasise the cell types founds. The infiltrate did appear to be mixed, however some immune-cell types predominated, as histiocytic, monocytic, and lymphocytic cell types. 

Eosinophilia in the gills? Please be more specific. Which cells show hypereosinophilia?

We observed an increased number of eosinophil granule cells in hyperplastic secondary lamellas,  however it does not appear to be consistent or linked to the CMF condition. An image is included in figure 2. 

Fouling of gills? Do authors mean autolysis?

We have altered the text to clarify. Fouling is used in this context to suggest an impact on gill health by particulate matter and environmental microbiota. Gill sections were obtained from recently deceased fish and fixed rapidly to prevent autolysis. There are some regions of gill tissue (particularly in the region of the branchial arch) that were less well preserved than others (as is routine with fish histopathology), however these were not to the extent or severity impair pathological assessments and did not require a comment. 

What is perivascular eosinophilia?

An increase of inflammatory cells, mainly eosinophils, around the blood vessels. We have changed it for the term perivasculitis.

Table 1: how many animals were examined? How many animals showed lesions? Please give numbers? Again, the terms in the table need to be explained, in the table or in the text, e.g. dermatitis – modifier, severity, etc.. Eosinophilia is not a patho term. What do authors mean? Lamina propria of which organ? Intestine?

The table has been altered to reflect the numbers of CMF fish in which specific pathological changes were observed. The unaffected fish sampled as part of the 2019 effort have been removed from the table to prevent confusion on the proportion of diseased fish with each clinical indicator of disease.  Whilst it was not possible to sample the entire population, and indeed the authors were reliant on the husbandry staff to select fish from the population. Randomisation of sampling cannot therefore be guaranteed, and so numbers, and protional findings are likely not representative of the population as a whole. Nevertheless, a contrast can be made between fish where no pathology was detected,

The description of dermatitis has been altered within the text to read “chronic dermatitis”. General comment is not made regarding severity of this inflammatory infiltration within tissue as this varied across fish examined, a subject for which an additional paragraph has been placed within text to address. Some fish were considered to be demonstrating a moderate dermatitis, present throughout the impacted region, and diffusely through microscopic field of view. Other fish, considered to be suffering a severe dermatitis with involvement of underlying tissues, are also described.

It did not seem appropriate to characterise the dermatitis further beyond describing it as chronic (other and provision of a detailed description) as the presentation was not considered sufficiently similar to other, previously described, dermal pathologies in salmonids. However we appreciate the feedback of the reviewers and their suggestion for inclusion of clear modifiers of the dermatitis in question.

Additional modifiers of the condition have now been included to highlight the diffuse natural of the cellular infiltration, with a clearer differentiation of what characterised a more severe change.

The lamina propria is of the stomach and this has been clarified now, we hope, within the text. The table has been altered to make it clear that the tissue in question is the stomach. 

Authors describe neutropilic infiltration in association with PKD. Neutrophils is not the prominent inflammatory cell with PKD. What did the authors see?

This section has been revised to clarify. Neutrophils were observed in the fish presenting the bacteria infection. PKD and nephrocalcinosis was also present with the tissue of this kidney showing inflammatory cell infiltration predominantly composed of neutrophils. We have clarified the text of the manuscript to resolve this confusion.

The cardiomyositis with the gram positive bacteria resembles very much BKD, an infection with Renibacterium salmoninarum. This should be clarified, as BKD is a chronic disease with a long incubation period of several months inducing granulomatous inflammation in several organs. This infection should be ruled out.

We have included a paragraph in the discussion to address this comment. Renibacterium salmoninarum infection can cause granulomas in several visceral organs. It can also cause multi-focal petechial hemorrhage around lateral line. However, the presentation of this skin rash  is different from the condition described in the present manuscript (for photos of the  “spawning rash” see Delghandi et al 2020).  We didn’t observe typical BKD clinical signs as skin darkening, distended abdomens, blood-filled blisters on the flanks, bruising around the vent, or internal granulomas in any of the CMF fish sampled.  This bacteria infection could be also a Carnobacterium. Some strains have been related with kidney disease and nephrocalcinosis, a comment has been included in the discussion. The low prevalence of the bacteria infection in the trouts showing CMF syndrome (3.3%) suggests a low correlation among pathologies. However, we will continue studying this pathology and any possible correlation with other infections. We appreciate the importance to ID this infection and have asked for internal funding for sequencing it from DNA extractions from the paraffin block.

Figure 2a: it would be worth to show a higher magnification as well. Please be more specific which inflammatory cells were involved. Figure 3: same, please specify which immune cells were involved.

Detail of inflammatory cells has been included in figure 4 and clarified in the text.

Figure 4 c-f can be deleted

We have deleted some pictures.  We however, believe that it is essential to keep at least a picture of PKD as this pathogen was highly prevalently found in the affected fish.  

Microbiology findings: first sentence belong to Materials and Methods

Sentence deleted.

Page 8, line 258: RNA instead of ARN

Corrected.

Host response: Authors examined immune gene expression in the kidney: was this correlated to infection with T. bryosalmonae? If fish were additionally infected with this parasite, immune gene expression is most probably related to the parasite infection instead of to the CMF syndrome.

The kidneys showed  pathologies as nephrocalcinosis and PKD in some of the CMF-fish included in the study.  These concurrent pathologies have impacted the observed host response in the kidneys pf CMF fish. This was already mentioned in the discussion in paragraphs 568-584.

Figure 7 can be deleted

Moved to the supplement.

Discussion

Page 13, line 390: what is a "stomach linen"? Serosa? Mucosa?

This typographical error has been remedied. The impacted regions include the lamina propria and muscular tissues.

Page 13, line 404: how can authors justify this statement? To confirm the absence of pathogen, a more thoroughly examination would be needed.

In the histopathology study, we could not identify any pathogen in the jaw lesions. Similarly, we could not isolate any bacteria or virus from the jaw sections. We plan to keep sampling more animals when the farm eventually returns to business.

Line 406: there are also infectious agents with an incubation time of several month, e.g. Renibacterium salmoninarum

The farm has no historical records of Renibacterium salmoninarum. We did not see gloss pathology characteristic of this infection in the multiple visits conducted by the FHI to the farm. We have never seen this pathology in other sites infected with R. salmoninarum. If this is an atypical presentation of R. salmoninarum, the bacteria infection and its characteristic granulomas should have been observed in the CMF fish. A paragraph discussing the possible involvement of  R. salmoninarum and other Gram-positive bacteria has been included.

The upregulation of the immune genes in the skin and kidney is not surprising as we see a mononuclear infiltration in the skin and an infection with T. bryosalmonae in the kidney. However, what is the interpretation of the results? Are there any indications for possible causes?

The skin gene expression indicated chronic fibrosis, with the up-regulation of antimicrobial peptides and immunoglobulin. Those biomarkers respond to extracellular parasites and exogenous molecules (ie toxins, contaminants). A Th1 response was not observed, so it is unlikely the involvement of a viral infection or other intracellular microorganisms.

In the kidney, the response suggested immune suppression (lines 569-584). We interpreted this as a response to PKD infection as myxozoans can promote immunosuppression to favour parasite development.  

Page 14, line 463: in my opinion, an association between the "new" syndrome and PKD is very questionable. More probably the PKD is associated with the sewage plant inlet without any association to the jaw changes.

We mentioned that the non seasonal presence of PKD could be attributed to the sewage plant. We do not know which involvement might have PKD in the CMF syndrome (if any). We have modified the sentence to point out that PKD and calcinosis were the most prevalent pathologies observed.

Materials and Methods

Page 15, line 503: Giemsa and gram stains are very superficial to rule out protists and bacteria. Especially, if the authors find Gram positive bacteria and do not go further!

The bacterial infection as observed only in a single fish. Tetracapsuloides bryosalmonae was identified in the histology and confirmed by Taqman qPCR. We did not see any evidence of other protist infections. No abnormalities were observed in blood smears.

Lines 521-525: why only go further in detail with IPN? Was there any suspicion for IPN?

Yes, IPNV is endemic in the rainbow tout population (around 20% of the population). As IPNV grows very fast in cell cultures it can mask other slow-growing viruses.

Virology and PCR: on which criteria were the viruses selected for further investigation? Did authors really expect these viruses to be involved in the jaw lesion? And if yes, why?

Virology and bacteriology analysis are part of our standard protocol in disease investigation, including histophatology and blood smears.  Different cell cultures are used to allow for the growth of a wide range of viruses. Some of the plates developed CPEs. We conducted specific PCRs for virus known to be pathogenic to rainbow trout. In the absence of positive amplification, we sequenced the cell supernatant and blasted it against the NCBI database and known viruses.  

NGS: it is pretty much sad that authors limited the NGS examination on the cell culture supernatants of CPE positive cell cultures. Why limiting the examination on viruses which grow on the cell lines?

We have tested in the laboratory the limit of detection of the MiSeq. We spiked host tissues with dilutions of a plasmid. The limit of detection of the MiSeq was 103 copies of the plasmid.  This limit of detection is acceptable when you have a virus growth on cells showing CPEs. However, for the direct sequencing of tissues where the ratio of host cells/pathogen is much higher, a deeper sequencing platform is required, as HiSeq. In this case, we would have needed to sequence at least 5 fish with lesions and 5 without to make it statistically robust. This analysis is not affordable for this kind of study, but we do not discard looking for funding if the condition spreads. This study aims to raise awareness of this condition. We regret that the causative agent was not found, but it is important to report these conditions as explained in the introduction.

Round 2

Reviewer 1 Report

Authors fairly addressed the reviewer comments.

Author Response

Dear reviewer,

Thanks for looking again at this manuscript. We have amended minor grammatical and improved the sentence structure. 

kind regards

Authors

Reviewer 3 Report

The manuscript has highly improved and most of the comments were considered. There are only few editorial changes left:

Page 4, lne 138: histiocytic and monocytic is more or less the same. Later in the mansucript, authors describe these cells as monocytic. To be consistent, please stay with this term.

Table 1 is a bit confusing. It is not clear which of the descriptions belong to which organ. A clearer presentation would help

Figure 2, line 173: please delete the (c)

Author Response

Dear Reviewer,

Thanks for your comments. We have amended grammatical errors and improved sentence structure.

Page 4, lne 138: histiocytic and monocytic is more or less the same. Later in the mansucript, authors describe these cells as monocytic. To be consistent, please stay with this term.

We have left “histiocytic” and removed ‘monocytic’, but we have left a mention to monocytes when referring to the circulation of blood cells.

Table 1 is a bit confusing. It is not clear which of the descriptions belong to which organ. A clearer presentation would help

We have amended the format of table 1 to make it clearer.

Figure 2, line 173: please delete the (c)

Thanks, we have corrected it.